# Cardiovascular Risk Factor Control in 70- to 95-Year-Old Individuals: Cross-Sectional Results from the Population-Based AugUR Study

**DOI:** 10.3390/jcm12062102

**Published:** 2023-03-07

**Authors:** Ferdinand J. Donhauser, Martina E. Zimmermann, Anna B. Steinkirchner, Simon Wiegrebe, Alexander Dietl, Caroline Brandl, Ralph Burkhardt, André Gessner, Frank Schweda, Tobias Bergler, Elke Schäffner, Carsten A. Böger, Florian Kronenberg, Andreas Luchner, Klaus J. Stark, Iris M. Heid

**Affiliations:** 1Department of Genetic Epidemiology, University of Regensburg, 93053 Regensburg, Germany; 2Statistical Consulting Unit StaBLab, Department of Statistics, LMU Munich, 80539 Munich, Germany; 3Department of Internal Medicine II, University Hospital Regensburg, 93053 Regensburg, Germany; 4Department of Ophthalmology, University Hospital Regensburg, 93053 Regensburg, Germany; 5Institute of Clinical Chemistry and Laboratory Medicine, University Hospital Regensburg, 93053 Regensburg, Germany; 6Institute of Clinical Microbiology and Hygiene, University Hospital Regensburg, 93053 Regensburg, Germany; 7Institute of Physiology, University of Regensburg, 93053 Regensburg, Germany; 8Department of Nephrology, University Hospital Regensburg, 93053 Regensburg, Germany; 9Institute of Public Health, Charité Universitätsmedizin Berlin, 10117 Berlin, Germany; 10Department of Nephrology, Hospital Traunstein, 83278 Traunstein, Germany; 11Institute of Genetic Epidemiology, Medical University of Innsbruck, 6020 Innsbruck, Austria; 12Department of Cardiology, Hospital Barmherzige Brüder Regensburg, 93049 Regensburg, Germany

**Keywords:** LDL-cholesterol, HbA1c, blood pressure, estimated glomerular filtration rate, urine albumin to creatinine ratio, elderly population

## Abstract

Cardiovascular risk factors such as high glucose, LDL-cholesterol, blood pressure, and impaired kidney function are particularly frequent in old-aged individuals. However, population-based data on the extent of cardiovascular risk factor control in the old-aged population is limited. AugUR is a cohort of the mobile “70+”-year-old population of/near Regensburg, recruited via population registries. We conducted cross-sectional analyses assessing the proportion of AugUR participants with LDL-cholesterol, HbA1c, or blood pressure beyond recommended levels and their association with impaired creatinine- and cystatin-based estimated glomerular filtration rate (eGFR, <60 mL/min/1.73 m^2^) or urine albumin–creatinine ratio (UACR, ≥30 mg/g). Among 2215 AugUR participants, 74.7% were taking lipid-, glucose-, blood-pressure-lowering, or diuretic medication. High LDL-cholesterol at ≥116 mg/dL was observed for 76.1% (51.1% among those with prior cardiovascular events). We found HbA1c ≥ 7.0% for 6.3%, and high or low systolic blood pressure for 6.8% or 26.5%, respectively (≥160, <120 mmHg). Logistic regression revealed (i) high HbA1c levels associated with increased risk for impaired kidney function among those untreated, (ii) high blood pressure with increased UACR, and (iii) low blood pressure with impaired eGFR, which was confined to individuals taking diuretics. Our results provide important insights into cardiovascular risk factor control in individuals aged 70–95 years, which are understudied in most population-based studies.

## 1. Introduction

The proportion of septuagenarians and octogenarians is constantly increasing in Western societies [1]. Moreover, the life expectancy of old-aged individuals is on the rise, which underlines the increased importance of the primary and secondary prevention of diseases, particularly cardiovascular disease, in the elderly. Common cardiovascular risk factors are increased blood pressure, LDL-cholesterol, and glucose concentrations. These risk factors are particularly often elevated among old-aged individuals [2,3,4]. Elevated levels are associated with an increased risk of cardiovascular disease, kidney damage, and increased mortality [5,6,7,8,9]. For old-aged individuals, also low blood pressure is associated with increased mortality [10,11,12]. 

There is a substantial debate on the old-aged individual’s benefit after lowering cholesterol levels [13,14], blood pressure [15,16], and HbA1c [17]. This debate can benefit from an understanding of the extent of cardiovascular risk factor control in the old aged. Several guidelines provide recommended levels to control LDL-cholesterol, HbA1c, and blood pressure, with partly different levels for the general population, for individuals at high cardiovascular risk, and for the old aged [7,8,9,18,19]. Medications to control these risk factors are the most commonly used drugs in the elderly [20]. Since impaired kidney function poses a substantial risk for cardiovascular events in itself [21], individuals with achieved lipid, glucose, or blood pressure control can be still at risk because of impaired kidney function. Unachieved control of LDL-cholesterol, HbA1c, and blood pressure is particularly relevant for individuals with concomitant low kidney function. Thus, a quantification of cardiovascular risk factor control should not lose sight of kidney function. 

Despite this debate, and the growing proportion of elderly in the population, observational data on the older population is scarce and the knowledge of the extent of cardiovascular risk factor control in old-aged individuals is limited [16,22,23]. This is related to challenges in conducting population-based studies in the elderly [24]. Including the old aged in population-based studies requires a study protocol and study program that is specifically tailored to their needs. Thus, most large-scale population-based studies exclude the old aged (e.g., UK Biobank and NAKO up to 69 years old [25,26]).

We thus aimed to understand the extent of cardiovascular risk factor control—with and without medication intake—in the elderly. We also aimed to provide a joint view with concurrent impaired kidney function: with low creatinine and cystatin-based eGFR, which assess impaired filtration, and with high UACR, which is a marker of kidney damage. For this, we conducted a cross-sectional analysis of 2215 participants of the AugUR study. AugUR is a population-based study of individuals aged 70 to 95 years from Regensburg, Germany. This included a detailed assessment of medication intake, medical history, blood pressure, HbA1c, LDL-cholesterol, estimated glomerular filtration rate (eGFR) based on creatinine as well as cystatin C, and urine albumin–creatinine ratio (UACR). Our specific aims were (i) to assess the taken medication; (ii) to quantify the proportion of individuals with LDL-cholesterol, HbA1c, and blood pressure beyond recommended levels, among those treated and untreated; and (iii) to test the association of unachieved risk factor control with concurrent kidney function impairment.

## 2. Materials and Methods

### 2.1. Study Design and Study Population

AugUR (Age-related diseases: understanding genetic and non-genetic influences—a study at the University of Regensburg) is a prospective cohort study, designed to understand the extent and determinants of common diseases in the elderly. Study design, protocols, and inclusion criteria were described in detail previously [27]. Briefly, a random sample of individuals aged at least 70 years, living in the area of Regensburg, a city of around 150,000 inhabitants in the south-east of Germany, was obtained from a population registry and contacted by mail. Among 13,522 individuals contactable by mail, 2449 individuals participated in the baseline assessment conducted in the years 2013 to 2019 (net response: 18.1%). Participants were required to reach the study center independently and answer all questions personally. Therefore, the AugUR participants were physically and mentally relatively healthy and reflected the mobile proportion of the old-aged population. The study program at the study center, the University Hospital Regensburg, included a standardized in-person interview, medical exams, as well as blood and urine collection. This work presents cross-sectional analyses using the AugUR baseline data.

The study was approved by the Ethics Committee of the University of Regensburg, Germany (vote 12-101-0258) and conducted according to the principles expressed in the Declaration of Helsinki. All study participants provided written consent after being informed about the study. 

### 2.2. Assessment of Medication Intake

AugUR participants were requested to take their medication packages/blisters and medication lists to the study center. Trained staff recorded all currently taken medication in the database. This database was linked with the Anatomical Therapeutic Chemical (ATC) classification to determine and record the active ingredient(s) [28]. For this work, three broad medication groups were defined as established previously [29]: (i) lipid-lowering agents (ATC group C10); (ii) glucose-lowering drugs (ATC group A10); (iii) blood-pressure-lowering drugs as any of the following—diuretics (except high-ceiling diuretics), beta blockers, angiotensin-converting enzyme inhibitors, angiotensin receptor blockers, renin inhibitors, calcium channel blockers, and other antihypertensives (ATC group C02). Refined subgroups according to active substances were defined where applicable. Plant, homeopathic, and anthroposophical substances were not considered.

### 2.3. Blood and Urine Biomarkers

Collection and processing of biosamples were conducted following standard operation procedures developed for this study based on established methods and recommendations [30], as described previously [27]. Briefly, non-fasting blood samples were drawn in a sitting position after at least 5 min of resting. Mild venous stasis was applied for a maximum duration of 1 min. Blood was taken using a 21G multifly needle. Immediate measurements in fresh whole blood and serum were carried out on the same day. Samples for biobanking were processed immediately and stored at −80 °C. Midstream urine was collected and directly stored at −80 °C. 

Measurements in fresh samples were carried out by an external laboratory (Synlab, Regensburg, Germany). HbA1c was measured from EDTA-anticoagulated whole blood by applying ion-exchange high-performance liquid chromatography on a Bio-Rad Variant II Turbo, applying the Variant II Turbo HbA1c Kit 2.0 (Bio-Rad, Munich, Germany). LDL-, HDL-, and total cholesterol were quantified as mg/dl from serum on a Beckman AU 5400 analyzer using enzymatic tests OSR6183, OSR6187, and OSR6116, respectively (Beckman Coulter, Krefeld, Germany).

Laboratory analyses from biobanked samples for creatinine, cystatin C, and albumin were performed in compliance with the “Guidelines of the German Medical Association for Quality Assurance of Medical Laboratory Tests” (RiLiBäK) at the Central Laboratory of the University Hospital Regensburg, which is accredited in accordance with the standard DIN EN ISO 15189. Creatinine from serum and midstream urine was enzymatically measured in individuals recruited in the years 2013–2015 (AugUR1, n = 1133) on a Siemens Dimension Vista 1500 (assay ECREA, Siemens Healthcare, Erlangen, Germany) or in those recruited from 2017 to 2019 (AugUR2, n = 1316) on a Roche cobas e801 (assay CREP2, Roche, Mannheim, Germany). Serum cystatin C was measured with immunoassays for AugUR1 on a Siemens Dimension Vista 1500 (assay CYSC) or for AugUR2 on a Roche cobas e801 (assay CYSC2). Urine albumin was measured with immunoassays for AugUR1 on a Siemens Dimension Vista 1500 (assay MALB) or for AugUR2 on a Roche cobas e801 (assay ALBT2). Comparability of methods for creatinine, cystatine C, and albumin was assessed following Clinical & Laboratory Standards Institute (CLSI) guidelines.

### 2.4. Assessment of Lifestyle Factors, Medical Conditions, and Chronic Diseases

At the study center, lifestyle factors and medical history were assessed in a standardized face-to-face interview. Specifically, participants were asked if they had ever been diagnosed by a physician with hypertension, diabetes, stroke, or heart failure. Additionally, they were asked about any history of myocardial infarction, stent implantation, or bypass surgery; coronary artery disease (CAD) was defined if at least one of these three conditions was reported. We built a variable that included CAD or stroke (CAD/stroke). Previous work has shown high agreement for the self-report of diabetes, stroke, CAD, and physician-reported comorbidities in AugUR, but limited reliability of self-reported heart failure [31]. Smoking status was defined as ever versus never smoking. 

Measurements at the study center included height, weight, and blood pressure. Systolic and diastolic blood pressure (SBP/DBP) was measured by an automatic device 3 times after >5 min resting, using the average of the second and third measurements in the analyses. Obesity was defined as body mass index (BMI) ≥ 30 kg/m^2^. Hypertension was defined as blood pressure ≥ 140/90 mmHg or if the individual reported a prior hypertension diagnosis and antihypertensive medication intake, as established previously [29]. Individuals who self-reported diabetes and/or antidiabetic medication intake were defined as diabetic [32]. Estimated glomerular filtration rate (eGFR) was assessed both creatinine-based and cystatin-based using the CKD-Epi equation [33,34]. Impaired glomerular filtration rate was defined as eGFR < 60 mL/min/1.73 m^2^, and albuminuria as UACR ≥ 30 mg/g [21]. Echocardiography was conducted in a subgroup of 796 participants. Ejection fraction was measured in the apical four-chamber view using Simpson’s method [35,36]. 

### 2.5. Statistical Analysis

We conducted cross-sectional analyses using the AugUR baseline data, including all participants with valid values for LDL-cholesterol, HbA1c, blood pressure, and medication intake. Continuous variables were reported as mean and standard deviation or as median and interquartile range. For categorical variables, percentages were reported. The distributions of cardiovascular risk factors were shown using box plots, separately for individuals with respective medication intake or without. 

We derived the proportion of participants at achieved cardiovascular risk factors as the proportion of individuals who had LDL-cholesterol, HbA1c, or blood pressure levels below recommended thresholds. For LDL-cholesterol, we considered the thresholds of the European Society of Cardiology and European Atherosclerosis Society (e.g., <116 mg/dL) [8]; for HbA1c, those of the American and the German Diabetes Association (<7.0%, ≤7.5%, respectively) [6,7]; for blood pressure, those of the European Society of Cardiology and the European Society of Hypertension (120–140/80–90 mmHg) [9]. Proportions of achieved levels were derived overall, by medication intake status, and separately for individuals with or without a prior diagnosis of CAD/stroke. In exploratory analyses, we tested whether women or men, old aged or very old aged (70–79, 80+), were more likely to have unachieved levels using logistic regression adjusting for CAD/stroke and, if applicable, diabetes (model I) and for respective medication intake (model II). 

We tested the association of unachieved levels with impaired kidney function: with creatinine- or cystatin-based eGFR < 60 mL/min/1.73 m^2^ or with UACR ≥ 30 mg/g. For this, we used multivariable logistic regression adjusted for age (continuous), sex, CAD/stroke, diabetes (if applicable), obesity, smoking, respective medication intake, and an interaction of unachieved levels with medication intake. Adjustment for CAD/stroke and diabetes was included to account for potential confounding by indication. No adjustment was made for heart failure, as self-reported heart failure is rather unreliable and the ejection fraction was only measured in a subgroup. In the sensitivity analyses, we applied a model without adjustment for CAD/stroke and diabetes and a model without obesity and smoking. 

The level of significance was set at *p* < 0.05, except for interaction terms (*p* < 0.1). RStudio for Windows, Version 1.4.1717, was used for the Loess function. Forest plots were designed with Microsoft Excel, Version 2022. For all other analyses, SPSS Statistics for Windows (IBM), Version 26.0, or R version 4.1.2 was used.

## 3. Results

### 3.1. Three Quarters of the Participants Aged 70 to 95 Years Were Taking Medication for Cardiovascular Risk Factor Control

Among the 2449 AugUR participants, we here analyzed the 2215 participants with valid values for LDL-cholesterol, HbA1c, blood pressure, and medication intake (Appendix A). These individuals were aged 70 to 95 years (mean = 78.4 years), 47.4% were men, 72.9% had hypertension, 21.0% diabetes, 15.0% CAD, 29.9% a creatinine-based eGFR < 60 mL/min/1.73 m^2^, 47.1% a cystatin-based eGFR < 60 mL/min/1.73 m^2^, and 17.4% had UACR ≥30 mg/g (Table 1).

Among the 2215 participants, 34.9% were taking lipid-lowering and 16.5% glucose-lowering medication (Table 2). This was similar for the old and the very old aged (n = 1469, 70–79 years; n = 746, 80+ years, respectively; Table 2), but lipid-lowering medication was less frequent among women than among men (29.5% versus 40.8%, respectively; Table 2). Blood-pressure-lowering medication was taken by 67.7%, mostly RAAS inhibitors (54.0% of the 2215); few were taking GLP1 analogues or SLGT2 inhibitors. High-ceiling diuretics were taken by 12.9%, with a marked increase among the very old aged versus old aged (18.6% versus 9.9%); further, 27.5% were taking other diuretics as part of antihypertensive therapy. Any of these medications were taken by 74.7%, and 25.3% were taking none of these. A characterization of individuals by treated versus untreated status is given in Appendix A.

### 3.2. LDL-, HbA1c, and Blood Pressure Levels Differed between Treated and Untreated

We compared quantitative risk factor levels by respective medication intake among the 2215 AugUR participants. We found (i) on average, lower LDL-cholesterol among treated compared to untreated (Figure 1A; mean = 118.4 mg/dL vs. 153.1 mg/dL, respectively; age- and sex-adjusted *p* < 0.001); (ii) markedly higher HbA1c among antidiabetic treated versus untreated (Figure 1B; mean = 6.73% vs. 5.60%, respectively, age- and sex-adjusted *p* < 0.001); and (iii) similar systolic blood pressure values between treated and untreated (Figure 1C,D; SBP: mean = 131.9 mmHg vs. 131.7 mmHg, age- and sex-adjusted *p* = 0.552; DBP: mean = 75.4 mmHg vs. 77.6 mmHg, age- and sex-adjusted *p* < 0.001). There were few differences between the old and the very old or between men and women (Appendix A).

### 3.3. Cardiovascular Risk Factor Control Was Partly Unachieved and Some Individuals Appeared Potentially Overtreated

We quantified the proportion of the 2215 participants who had cardiovascular risk factors above recommended levels, indicating unachieved risk factor control. This yielded a diverse pattern (Table 3). (i) LDL-cholesterol control was rather poor—few reached the recommended thresholds at <70 mg/dL or <100 mg/dL (0.9%, 12.1%, respectively), while most (76.1%) had high LDL-cholesterol levels ≥116 mg/dL [8]. Among individuals with lipid-lowering medication (n = 772, 322 of these with previous CAD or stroke), values at <116 mg/dL were achieved by 50.3%. This was similar for individuals with previous CAD or stroke (n = 473), who were mostly treated. However, 151 individuals with a previous CAD or stroke diagnosis reported no lipid-lowering medication intake and 83.4% of these had values ≥116 mg/dL.

(ii) HbA1c control was excellent; only 6.3% had HbA1c ≥ 7.0% [6]. However, 71 individuals treated with antidiabetic medication had levels <6.0% (19.5%), considered too low by the American Diabetes Association [6]. Among the 1850 untreated individuals, n = 14 (0.8%) had HbA1c ≥ 7.0%, which might indicate undetected diabetes.

(iii) High SBP at ≥160 mmHg or DBP at ≥100 mmHg was rare (6.8% or 2.3%, respectively). However, a large number of individuals were at low levels (<120 mmHg or <80 mmHg) considered undesirable for old-aged individuals: 26.5% had SBP <120 and 64.8% DBP <80 mmHg. These low levels were particularly frequent among individuals taking diuretics (31.7% and 72.3% among the n = 802). One may be interested in whether individuals with a prior CAD or stroke diagnosis were similarly or even better controlled for high blood pressure: this was rather similar (94.1% and 98.5% at SBP < 160 or DBP < 100, respectively).

In order to identify potential disparities in the proportion of unachieved levels, we evaluated the association of sex, age group (80+, 70–19 years), and their interaction on the risk of unachieved control. For this, we used logistic regression adjusted for CAD/stroke and diabetes (if applicable). We found that (i) women were more likely to be at unachieved LDL-cholesterol levels or too low blood pressure, and (ii) old men more likely at unachieved HbA1c or too high blood pressure levels compared to women or very old men (Appendix A). This was not mediated by a prior diagnosis of CAD/stroke or diabetes diagnosis (since this was adjusted for), nor a differential probability of treatment (further model adjusting for treatment, Appendix A). However, the dosages of medication intake were not ascertained here, and differential dosages or differential impacts of similar dosages might explain at least some of these differences.

### 3.4. Regarding the Cross-Sectional Association of Unachieved Cardiovascular Risk Factor Control with Impaired Kidney Function

We evaluated the cross-sectional association of unachieved cardiovascular risk factor control with eGFR < 60 mL/min/1.73 m^2^ and UACR ≥ 30 mg/g. We used logistic regression adjusted for age, sex, prior diagnosis of CAD/stroke, diabetes (if applicable), obesity, smoking, the respective medication intake, and its interaction with risk factor control. In the sensitivity analyses, we applied a model without adjustment for CAD/stroke and diabetes and another model without adjustment for obesity and smoking, both yielding similar results (Appendix A). For blood pressure, we tested high as well as low blood pressure (SBP ≥ 140 vs. 120–140 mmHg or DBP ≥ 90 vs. 80–90 mmHg; SBP < 120 mmHg vs. 120–140 mmHg or DBP < 80 mmHg vs. 80–90 mmHg). 

We found LDL-cholesterol levels ≥116 mg/dL to be not associated with the risk for impaired creatinine-based eGFR, but with a decreased risk for impaired UACR (OR = 0.754, *p* = 0.049; Figure 2, Appendix A), without interaction by treatment status. 

We found unachieved HbA1c levels (e.g., ≥7.0%) significantly associated with a ~3-fold increased odds for impaired creatinine-based eGFR among the participants untreated for diabetes (OR = 3.075, *p* = 0.044; Figure 2, Appendix A). The same tendency was observed for impaired cystatin-based eGFR and UACR, but was not significant. Of note, the number of individuals that were untreated for diabetes and had high HbA1c (≥7.0%) were few (n = 14). Nevertheless, for these few individuals, this can be an important finding. 

High blood pressure (SBP ≥ 140 or DBP ≥ 90 mmHg) showed a tendency of decreased risk for impaired eGFR (e.g., creatinine-based: OR = 0.788, *p* = 0.051 or OR = 0.691, *p* = 0.076, respectively) and a significantly increased risk for impaired UACR (OR = 1.305, *p* = 0.049 or OR = 2.122, *p* < 0.001, respectively; Figure 2, Appendix A). There was no interaction with antihypertensive therapy intake. 

Low blood pressure (SBP < 120 or DBP < 80 mmHg) was not associated with impaired UACR (OR = 0.760, *p* = 0.078 or OR = 0.883, *p* = 0.391, respectively). However, it was associated with an increased risk for impaired eGFR (low SBP with creatinine-based eGFR: OR = 1.274, *p* = 0.045; low SBP and low DBP with cystatin-based eGFR: OR = 1.509, *p* = 0.001 or OR = 1.383, *p* = 0.005, respectively). For the association of low SBP and creatinine-based eGFR, we found a significant interaction with diuretic treatment (*p*-interaction = 0.008) and this association was predominantly among individuals taking diuretics. We visualized this cross-sectional finding between low SBP and impaired eGFR for quantitative levels using Loess splines (i.e., no linearity assumption; Figure 3): this substantiated again that the low blood pressure levels concomitant with low eGFR were mostly observed for individuals on diuretics, particularly when SBP was lower than 110 mmHg. Individuals at SBP < 110 mmHg with concomitant eGFR < 60 mL/min/1.73 m^2^ were undergoing particularly intense antihypertensive/diuretic therapy—with regard to the number of different agents or intake of high-ceiling diuretics (>2 antihypertensive agents: 41.1%; high-ceiling diuretics: 43.3%; Appendix A).

## 4. Discussion

Our cross-sectional study of 2215 individuals aged 70–95 years provides important insights into the extent of medication intake and cardiovascular control with or without medication in the elderly in a German population. We found partly well-achieved and partly unachieved cardiovascular risk factor control in an old-aged mobile community-dwelling population. Risk factor control was poor for LDL-cholesterol, excellent for HbA1c, and mixed for blood pressure. We obtained evidence for potential under- and overtreatment, with some potential disparities by sex and age groups. 

We also observed a complex pattern of association with kidney function: elevated LDL-cholesterol showed a counter-intuitive association with a decreased risk of kidney damage irrespective of treatment status. High HbA1c among individuals without antidiabetic therapy was associated with an increased risk of impaired kidney function. Too high or too low blood pressure values were associated with an increased risk for kidney damage or impaired filtration, respectively. By this, we found several lines of evidence that complemented previous data, and we contribute with insights into the interaction with treatment: whether kidney function is impaired irrespective of treatment, only among those that are untreated, or only among those that are treated. The first appeared to be the case for high LDL-cholesterol and the inverse association with kidney damage irrespective of treatment status, the second for individuals at high HbA1c without antidiabetic therapy associated with impaired kidney function, and the third for individuals with low blood pressure and simultaneously low eGFR, observed predominantly among individuals on diuretics. 

However, we are well aware that the cross-sectional, observational nature of our data does not allow for a clinical judgement on the best therapy, nor for a causal link between cardiovascular risk factors or therapy and impaired kidney function. Selection and indication bias need to be considered. To this end, it should be noted that our study sample reflects diseases and conditions proportional to a “mobile” elderly population, since participants were able to come to the study center and answer all questions personally [27]. Furthermore, our results are adjusted for a history of CAD/stroke to account for confounding by indication, but not for self-reported heart failure due to discrepancies with physician reports [31]. We discuss our findings more specifically in the following. 

LDL-cholesterol control at 116 mg/dl was unachieved by 76% [8]. It is unclear whether this should be considered undertreatment, since there is an ongoing debate on the benefit of lipid control among the old aged [8,18,37]. However, still, 51% of individuals with a prior CAD or stroke had LDL-cholesterol ≥ 116 mg/dL, which place them at high risk for further cardiovascular events under the “cholesterol hypothesis”. This risk is viewed as being reducible by lowering LDL-cholesterol, even at old age [8]. Nevertheless, there is substantial uncertainty in how to judge high LDL-cholesterol in the elderly: a lack of or an inverse association of LDL-cholesterol with mortality was observed in the old aged [38], with potential reasons being “inverse causation”, the beneficial effects of high LDL-cholesterol in the elderly, or the adverse effects of treatment. Our finding of high LDL-cholesterol associated with a decreased risk of kidney damage was in line with this inverse association in the elderly. That this was irrespective of treatment status might suggest other explanations than adverse treatment effects. 

Only 6.3% had HbA1c ≥ 7.0% and 2.8% had HbA1c > 7.5% [6,7], which indicates excellent glucose control. Nonetheless, among the individuals without antidiabetic therapy, we observed 14 individuals with HbA1c ≥ 7.0% and HbA1c ≥ 7.0% significantly associated with impaired filtration, and a tendency also for increased risk of kidney damage. Among the individuals with antidiabetic therapy, we observed 71 individuals with HbA1c at <6.0%; this is considered too low according to guidelines due to a high risk for hypoglycemia [6]. Our data thus suggest some individuals with untreated diabetes possibly at an increased risk for impaired kidney function, and some individuals with potential overtreatment. However, clinical routine data would be required for a definitive judgement.

Very high systolic or diastolic blood pressure at ≥160 or ≥100 mmHg was found for 7% or 2% of individuals, respectively. There is a broad consensus that such levels are considered too high also at old age. Thus, these individuals can be considered undertreated [9,18]. We found high blood pressure not associated with impaired filtration, but with a significantly increased risk with UACR ≥30 mg/g. Both were in line with the Berlin Initiative study [39] (n = 2069, aged 70+ years). This was also in line with the Leiden 85-Plus Study [40] (n = 550, aged 85+ years), which reports a lack of high blood pressure association with creatinine clearance, but without data on UACR. 

The benefit of lowering to 140/90 mmHg or lower for the old aged is controversially discussed [10,11,18,41]. More than a quarter of AugUR participants had low blood pressure at <120/80 mmHg. Such low blood pressure might place old-aged individuals at an increased risk for falls and mortality [9,12,42]. However, the SPRINT trial provided evidence that a systolic blood pressure target at <120 mmHg compared to <140 mmHg reduced all-cause mortality in old-aged individuals [41]. Notably, SPRINT focusses on the very healthy old aged, e.g., it excludes individuals with diabetes or a history of stroke (compared to AugUR: diabetes 21%, history of stroke 9%). Regarding kidney function, SPRINT showed that individuals allocated to the lower blood pressure target suffered more likely from a reduction in eGFR. This is in line with our cross-sectional results that low blood pressure was associated with a higher risk of impaired eGFR, which confirmed previous findings on lower creatinine clearance in the presence of low blood pressure in old-aged individuals (Leiden 85-Plus Study [40]). Our data extend the previous findings in two ways: we observed low blood pressure as not associated with an increased risk of impaired UACR, but rather with a tendency towards a decreased risk. This suggests that the association between low blood pressure and impaired eGFR might not indicate structural kidney damage, but rather reduced perfusion pressure [9,43,44,45]. Furthermore, our interaction analyses showed the association of low systolic blood pressure with impaired eGFR to be predominantly pronounced among individuals taking diuretics. While we accounted for previous CAD or stroke and thus for potential confounding by indication, we did not attempt adjustment for heart failure, since self-reported heart failure is unreliable [31]. Whether this intensive therapy was warranted, e.g., due to heart failure or a potential overtreatment, cannot be judged from these cross-sectional data.

Our study comprises a relatively large sample of elderly individuals with standardized assessments of cardiovascular risk factors, which fills a gap in the current literature. A strength of our study is the evaluation of taken medication, rather than relying on information about prescribed medication. The assessment of creatinine- and cystatin-based eGFR as well as UACR enables a view on kidney function beyond pure creatinine metabolism, as well as on structural kidney damage. Self-reported history of CAD, stroke, or diabetes has been shown to successfully reflect physician diagnoses [31]. Blood pressure was measured three times in the study center and the mean of the second and third measurements was used for analysis to minimize white-coat effects [27]. AugUR is designed as a longitudinal study, which will yield follow-up information in the future. Some limitations warrant mentioning. Our analysis is cross-sectional, which limits the interpretation regarding the sequence of events; we can show the co-occurrence of adverse cardiovascular risk factors and impaired kidney function, but longitudinal data or even a randomization to differential treatment regimen are warranted to enable a better judgement of over- or undertreatment and the respective consequences for cardiovascular risk factors and kidney function. Since the response to study invitations was ~18% and all participants had to come to the study center and answer all questions personally, AugUR captures a physically mobile and mentally healthy population over 70 years of age with an interest in health questions.

## 5. Conclusions

In summary, our data provide important insights into the extent of cardiovascular risk factor control in individuals at 70 to 95 years of age. Our results showed that recommended target levels were partly well achieved, but also partly unachieved in the elderly, indicating some potential undertreatment. Our data also suggest hypoglycemia or too low blood pressure by overtreatment in a few individuals. We thus provide data that may foster the debate on whether old-aged individuals would benefit from more or less therapy. Our results also provide an understanding of the cross-sectional association of cardiovascular risk factors with impaired kidney function. Elevated LDL-cholesterol showed a counter-intuitive association with a decreased risk of kidney damage, an inverse association typically seen for the elderly, with reasons being elusive. High HbA1c among individuals without antidiabetic therapy associated with impaired kidney function suggested some undetected diabetes with potential involvement of the kidney. Too high or too low blood pressure values were associated with an increased risk for kidney damage or impaired filtration, respectively. Low levels of blood pressure simultaneously with a low filtration rate were mostly noted among individuals taking diuretics. 

## Figures and Tables

**Figure 1 jcm-12-02102-f001:**
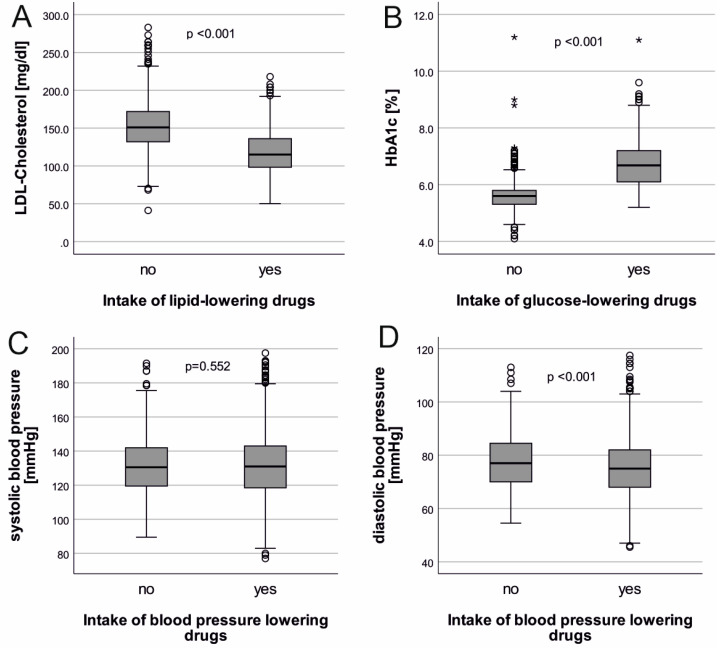
Distribution of LDL-cholesterol, HbA1c, systolic and diastolic blood pressure stratified by respective medication. Shown is the distribution of LDL-cholesterol stratified by intake of lipid-lowering drugs (**A**), of HbA1c stratified by intake of glucose-lowering drugs (**B**), and of systolic as well as diastolic blood pressure stratified by intake of antihypertensive drugs (**C**,**D**). Analyzed were 2215 AugUR participants. Shown are median, 25th and 75th percentiles (box), upper and lower whiskers (±1.5 IQR), as well as outliers beyond ±3 IQR as asterisks. Age- and sex-adjusted *p*-values from linear regression are given at the top of each panel, which tested the difference in the target parameter between participants with and without respective medication.

**Figure 2 jcm-12-02102-f002:**
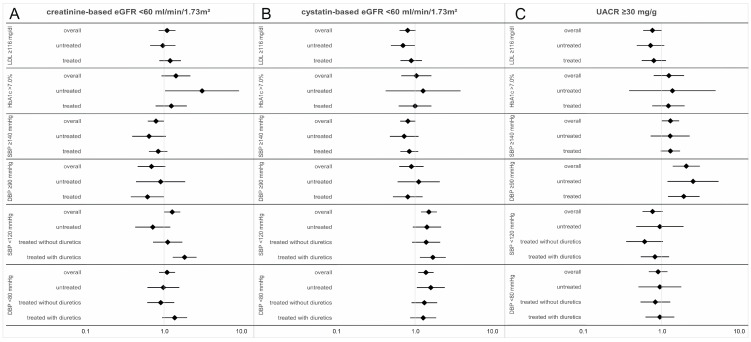
Association of unachieved cardiovascular risk factor control with impaired kidney function. Shown are odds ratios and 95% confidence intervals for the association of unachieved risk factor control with creatinine- or cystatin-based eGFR < 60 mL/min/1.73 m^2^ (**A**,**B**) and with UACR ≥ 30 mg/g (**C**). We applied logistic regression adjusted for age, sex, CAD/stroke, diabetes (if applicable), obesity (BMI ≥ 30 vs. <30 kg/m^2^), smoking (ever vs. never), and respective medication intake without interaction (overall model). We also applied a model adding an interaction between treatment and risk factor control, ~b0 + b1age + b2sex + b3CAD/stroke + b4diabetes + b5obesity + b6smoking + b7treatment + b8control + b9treatment * control; we derived the ORs for treated individuals by exp(b8 + b9) with corresponding 95% CIs (details in Appendix A). For low SBP/DBP, we used two variables for “treatment” (“treated without diuretics”: 1 = antihypertensives without diuretics, 0 = otherwise; “treated with diuretics”: 1 = diuretics, 0 = otherwise) and respective interaction parameters.

**Figure 3 jcm-12-02102-f003:**
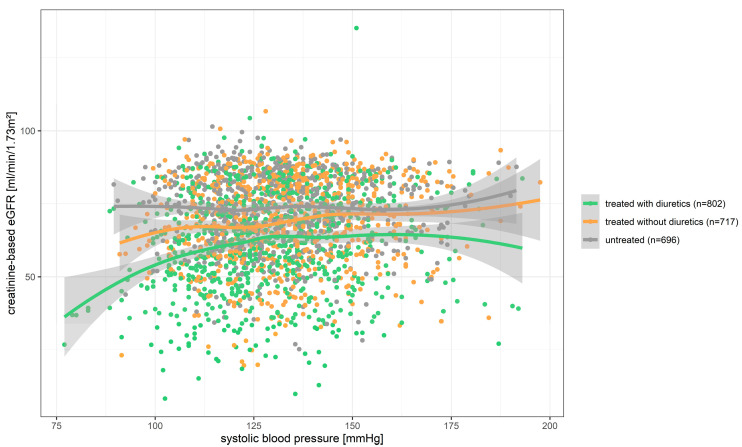
Creatinine-based eGFR versus systolic blood pressure stratified by blood-pressure-lowering medication and diuretics intake. Shown are values for each of the 2215 AugUR participants. A stratification was made between individuals without antihypertensive medication (grey), those with antihypertensive medication but without diuretic drugs (orange), and those taking diuretics (green). Loess best-fit curves with 95% confidence intervals were modeled for each group.

**Table 1 jcm-12-02102-t001:** Participant characteristics. The dataset was restricted to individuals with valid values for LDL-cholesterol, HbA1c, blood pressure, and medication intake. Shown are descriptive characteristics of the analyzed AugUR participants (n = 2215). Proportions (%) and numbers or means/median plus standard deviations (SD) or interquartile ration (IQR) are given.

Characteristics (n with Available Data, if <2215)	Overall (n = 2215)	70–79 yrs. (n = 1469)	≥80 yrs. (n = 746)
Men (% (n))	47.4% (1051)	48.1% (706)	46.2% (345)
Age [yrs] (mean (±SD))	78.4 (±5.0)	75.4 (±2.6)	84.2 (±3.4)
BMI [kg/m^2^] (mean (±SD))	27.7 (±4.5)	27.7 (±4.6)	27.6 (±4.3)
BMI ≥ 30 kg/m^2^ (% (n))	26.0% (575)	26.0% (382)	25.9% (193)
Smoking (ever) (% (n))	44.6% (988)	47.9% (704)	38.1% (284)
**Cardiovascular diseases**			
Stroke (% (n)) (n = 2207)	8.7% (191)	8.0% (117)	9.9% (74)
CAD ^a^ (% (n)) (n = 2208)	15.0% (332)	12.5% (184)	19.8% (148)
Self-reported heart failure (% (n)) (n = 2196)	15.1% (331)	12.6% (184)	19.9% (147)
Ejection fraction ^b^ [%] (mean (±SD)) (n = 796)	60.2 (±7.7)	60.3 (±7.5)	60.2 (±8.2)
Ejection fraction ^b^ ≤40% (% (n)) (n = 796)	1.6% (13)	1.6% (9)	1.7% (4)
**Lipids**			
Total cholesterol [mg/dL] (mean (±SD))	217.7 (±45.9)	219.0 (±45.6)	215 (±46.3)
HDL-cholesterol [mg/dL] (mean (±SD))	61.2 (±15.5)	61.1 (±15.5)	61.5 (±15.5)
LDL-cholesterol [mg/dL] (mean (±SD))	141.0 (±34.8)	142.2 (±34.5)	138.6 (±35.2)
**Diabetes mellitus**			
Diabetes ^c^ (% (n))	21.0% (466)	19.8% (291)	23.5% (175)
HbA1c [%] (mean (±SD))	5.79 (±0.68)	5.76 (±0.67)	5.83 (±0.70)
**Blood pressure**			
Hypertension ^d^ (% (n))	72.9% (1614)	70.3% (1033)	77.9% (581)
Systolic blood pressure [mmHg] (mean (±SD))	131.6 (±18.0)	131.5 (±17.7)	131.8 (±18.4)
Diastolic blood pressure [mmHg] (mean (±SD))	76.1 (±10.6)	77.1 (±10.4)	74.1 (±10.8)
**Kidney function**			
eGFRcrea ^e^ [mL/min/1.73 m^2^] (mean (±SD))	67.7 (±16.1)	70.6 (±15.3)	62.1 (±16.1)
eGFRcrea ^e^ <60 mL/min/1.73 m^2^ (% (n))	29.9% (663)	23.0% (338)	43.6% (325)
eGFRcys ^f^ [mL/min/1.73 m^2^] (mean (±SD)) (n = 2208)	60.7 (±16.8)	64.6 (±16.1)	52.8 (±15.2)
eGFRcys ^f^ <60 mL/min/1.73 m^2^ (% (n)) (n = 2208)	47.1% (1043)	36.2% (530)	69.1% (513)
UACR [mg/g] (median (IQR)) (n = 2150)	12.9 (7.1–27-4)	12.0 (6.7–24.1)	16.0 (8.1–36.1)
UACR ≥ 30 mg/g (% (n)) (n = 2150)	17.4% (375)	14.9% (214)	22.5% (161)
UACR > 300 mg/g (% (n)) (n = 2150)	2.9% (62)	2.4% (35)	3.8% (27)

CAD = coronary artery disease; eGFRcrea, eGFRcys = estimated glomerular filtration rate calculated from serum creatinine or serum cystatin C, respectively; UACR = urine albumin–creatinine ratio; (^a^) self-reported history of myocardial infarction, bypass surgery, and/or stent implantation; (^b^) measured in four-chamber view using Simpson´s method [35,36]; (^c^) self-reported diabetes and/or antidiabetic treatment [32]; (^d^) measured blood pressure ≥ 140/90 mmHg or (antihypertensive treatment and positive self-report) [29]; (^e^) via CKD-Epi formula [33]; (^f^) via CKD-Epi formula [34].

**Table 2 jcm-12-02102-t002:** Medication intake. Shown is medication intake of AugUR participants grouped by ATC coding. Frequencies are reported as proportion (%, relative to sample size n = 2215) and numbers, as well as stratified by age group and sex.

Medication Intake (% (n))	Overall (n = 2215)	70–79 yrs. (n = 1469)	≥80 yrs. (n = 746)	Women (n = 1164)	Men (n = 1051)
**Lipid-lowering**	34.9% (772)	35.1% (515)	34.5% (257)	29.5% (343)	40.8% (429)
Statin	34.1% (755)	34.0% (500)	34.2% (255)	28.7% (334)	40.1% (421)
Fibrate	0.6% (14)	0.8% (12)	0.3% (2)	0.8% (9)	0.5% (5)
Other	2.8% (62)	2.9% (43)	2.5% (19)	1.5% (18)	4.2% (44)
**Glucose lowering**	16.5% (365)	16.4% (241)	16.6% (124)	14.3% (167)	18.8% (198)
Insulin	4.2% (93)	4.1% (60)	4.4% (33)	3.5% (41)	4.9% (52)
Biguanide	11.0% (244)	11.5% (169)	10.1% (75)	9.9% (115)	12.3% (129)
Sulfonylureas	2.8% (61)	2.5% (37)	3.2% (24)	2.2% (26)	3.3% (35)
DDP4 inhibitors	3.8% (85)	4.0% (59)	3.5% (26)	3.0% (35)	4.8% (50)
Alpha glucosidase inhibitors	0.3% (7)	0.3% (4)	0.4% (3)	0.3% (3)	0.4% (4)
GLP1 analogues	0.2% (4)	0.3% (4)	0.0% (0)	0.3% (3)	0.1% (1)
SGLT2 inhibtors	0.8% (17)	0.9% (13)	0.5% (4)	0.6% (7)	1.0% (10)
Glinides	0.7% (15)	0.8% (12)	0.4 (3)	0.4% (5)	1.0% (10)
**Blood pressure lowering**	67.7% (1499)	64.7% (950)	73.6% (549)	67.5% (786)	67.8% (713)
Diuretics ^a^	27.5% (610)	25.6% (376)	31.4% (234)	27.5% (320)	27.6% (290)
Beta-blockers	27.0% (598)	25.2% (370)	30.6% (228)	26.8% (312)	27.2% (286)
Calcium channel blockers	23.5% (521)	21.6% (317)	27.3% (204)	23.6% (275)	23.4% (246)
RAAS inhibitors	54.0% (1197)	51.6% (758)	58.8% (439)	53.4% (621)	54.8% (576)
ACE inhibitors	26.0% (575)	24.8% (364)	28.3% (211)	23.1% (269)	29.1% (306)
Angiotensin receptor blockers	28.0% (620)	26.7% (392)	30.6% (228)	30.2% (352)	25.5% (268)
Renin inhibitors	0.2% (4)	0.2% (3)	0.1% (1)	0.1% (1)	0.3% (3)
Other	2.2% (48)	2.1% (31)	2.3% (17)	1.9% (22)	2.5% (26)
**Diuretics**	36.2% (802)	32.5% (477)	43.6 (325)	34.8% (405)	37.8% (397)
Low-ceiling, thiazides	24.2% (536)	22.7% (333)	27.2% (203)	24.3% (283)	24.1% (253)
Low-ceiling, excl. thiazides	2.0% (45)	1.7% (25)	2.7% (20)	1.5% (18)	2.6% (27)
High-ceiling	12.9% (285)	9.9% (146)	18.6% (139)	11.3% (131)	14.7% (154)
Potassium-sparing agents	4.7% (103)	4.2% (62)	5.5% (41)	4.0% (46)	5.4% (57)
Any of the above	74.7% (1655)	72.0% (1057)	80.2% (598)	73.5% (855)	76.1% (800)
None of the above	25.3% (560)	28.0% (412)	19.8% (148)	26.5% (309)	23.9% (251)

(^a^) Diuretics without high-ceiling diuretics.

**Table 3 jcm-12-02102-t003:** Proportion of achieved and unachieved cardiovascular risk factor control. Shown is the proportion of the 2215 AugUR participants at recommended LDL-cholesterol, HbA1c, or blood pressure levels, overall and separately by individuals with or without the respective treatment. With regard to LDL-cholesterol, recommended levels of the European Society of Cardiology and Atherosclerosis (ESC/EAS) are presented [8]. For HbA1c, levels at <7.0% or ≤7.5% are recommended by the American and German Diabetes Association [6,7]; levels <6.0% can be a sign of hypoglycemia for diabetes patients [6]. For blood pressure, systolic and diastolic blood pressure ranges defined by the European Society of Cardiology and Hypertension are reported [9]. Shown are also the proportions of participants at recommended levels among the 473 individuals with a prior diagnosis of CAD or stroke. These individuals are at high risk for cardiovascular events, where stricter medication and stricter goals are recommended.

LDL-Cholesterol	Overall				Participants with History of CAD ^a^ or Stroke			
	Overall (n = 2215)	Untreated (n = 1443)	Treated (n = 772)		Overall (n = 473)	Untreated (n = 151)	Treated (n = 322)	
<40 mg/dL	0 (0)	0 (0)	0 (0)		0 (0)	0 (0)	0 (0)	
<55 mg/dL	0.1% (3)	0.1% (1)	0.3% (2)		0.4% (2)	0 (0)	0.6% (2)	
<70 mg/dL	0.9% (20)	0.1% (2)	2.3% (18)		3.2% (15)	0 (0)	4.7% (15)	
<100 mg/dL	12.1% (268)	3.7% (54)	27.7% (214)		29.2% (138)	4.6% (7)	40.7% (131)	
<116 mg/dL	25.1% (557)	11.7% (169)	50.3% (388)		49.0% (232)	16.6% (25)	64.3% (207)	
≥116 mg/dL	76.1% (1685)	88.3% (1274)	49.7% (384)		51.0% (241)	83.4% (126)	35.7% (115)	
**HbA1c**	**Overall** **(n = 2215)**	**Untreated (n = 1850)**	**Treated (n = 365)**		**Overall (n = 473)**	**Untreated (n = 341)**	**Treated (n = 132)**	
<6.0%	74.0% (1640)	84.8% (1569)	19.5% (71)		59.0% (279)	76.2% (260)	14.4% (19)	
6.0–7.0%	19.7% (436)	14.4% (267)	46.3% (169)		28.8% (136)	22.0% (151)	46.2% (61)	
<7.0%	93.7% (2076)	99.2% (1836)	65.8% (240)		87.7% (415)	98.2% (335)	60.6% (80)	
≤7.5%	97.2% (2154)	99.8% (1847)	84.1% (307)		95.6% (452)	99.7% (340)	84.8% (112)	
**Systolic blood pressure**	**Overall (n = 2215)**	**Untreated (n = 696)**	**Treated without diuretics (n = 717)**	**Treated with diuretics (n = 802)**	**Overall (n = 473)**	**Untreated (n = 60)**	**Treated without diuretics (n = 159)**	**Treated with diuretics (n = 254)**
<100 mmHg	2.2% (49)	1.4% (10)	1.5% (11)	3.5% (28)	3.6% (17)	5.0% (3)	3.8% (6)	3.1% (8)
100–119 mmHg	24.3% (539)	23.7% (165)	20.6% (148)	28.2% (226)	27.5% (130)	21.7% (13)	25.8% (41)	29.9% (76)
120–139 mmHg	43.1% (955)	45.5% (317)	44.5% (319)	39.8% (319)	39.7% (188)	46.7% (28)	42.1% (67)	36.6% (93)
140–159 mmHg	23.5% (521)	22.6% (157)	24.8% (178)	23.2% (186)	23.3% (110)	18.3% (11)	23.3% (37)	24.4% (62)
≥160 mmHg	6.8% (151)	6.8% (47)	8.5% (61)	5.4% (43)	5.9% (28)	8.3% (5)	5.0% (8)	5.9% (15)
**Diastolic blood pressure**	**Overall (n = 2215)**	**Untreated (n = 696)**	**Treated without** **diuretics** **(n = 717)**	**Treated with diuretics (n = 802)**	**Overall (n = 473)**	**Untreated (n = 60)**	**Treated without diuretics (n = 159)**	**Treated with diuretics (n = 254)**
<70 mmHg	28.8% (639)	22.6% (157)	24.1% (173)	38.5% (309)	39.7% (188)	25.0% (15)	36.5% (58)	45.3% (115)
70–79 mmHg	36.0% (797)	37.4% (260)	37.1% (266)	33.8% (271)	36.2% (171)	41.7% (25)	42.1% (67)	31.1% (79)
80–89 mmHg	25.1% (557)	28.7% (200)	26.6% (191)	20.7% (166)	17.8% (84)	23.3% (14)	14.5% (23)	18.5% (47)
90–99 mmHg	7.7% (171)	8.9% (62)	9.3% (67)	5.2% (42)	4.9% (23)	10.0% (6)	5.0% (8)	3.5% (9)
≥100 mmHg	2.3% (51)	2.4% (17)	2.8% (20)	1.7% (14)	1.5% (7)	0 (0)	1.9% (3)	1.6% (4)

CAD = coronary artery disease; (^a^) self-reported history of myocardial infarction, bypass surgery, and/or stent implantation.

## Data Availability

Data are available upon reasonable request. The data include sensitive human person-specific information and are thus subject to data protection.

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
