# Peer review of "Cardiovascular Risk Factor Control in 70- to 95-Year-Old Individuals: Cross-Sectional Results from the Population-Based AugUR Study"

_jcm, 2023, doi:10.3390/jcm12062102_

Round 1
Reviewer 1 Report
The aim of the proposed paper is to assess to what extent 3 cardiovascular risk factors (CVRF) are well-controlled by treatment. To do so, they independently investigate whether relevant CVRF-related measures are within recommended ranges with respect to patients being treated or not.
Besides, they investigate the association between unachieved CVRF control and kidney function. The rationale for studying kidney function seems to be related to the fact that impaired kidney function is another high cardiovascular risk factor, meaning that a causal link might be hypothesised, although not properly introduced (see comment below).
Overall, the study is well designed and conducted. The collected data is consistent with the addressed question. Results are of great interest in terms of epidemiology as well as patient care.
My main concern is about CVRF interactions. Although it is sound to independently investigate the measured CVRFs (hypertension, glycemia, cholesterol), it is crucial to take into account overweight and smoking at least as moderating or confounding factors (overweight related to glycemia and smoking related to hypertension). They are surprisingly completely overlooked in the paper, although well documented.
The statistical analyses must be revised to take into account these factors.
Besides, I found the rationale behind forming age groups (80 years cutoff) rather arbitrary. Maybe some analyses using age as a continuous variables should be more appropriate? Moreover, group sizes are unbalanced. Since the study is not designed to stratify by age, this seems to be an exploratory analysis and should be presented as such. That said, the question related to age seems to be out of the scope of the paper, Indeed, they focus on "old aged" population (>69 years) as opposed to most studies that are restricted to less old people. The "old aged" is a sufficiently interesting group. I would advise to remove the age-related analyses and give more room and focus on CVRF interactions with respect to treatment, that is the major contribution of this paper.
My last comment is about kidney function impairment which is not properly introduced. There is only one sentence in the introduction about it although it is presented as a specific aim of the study. Why is it important to study this function concurrently with the other CVRFs? Is there a causal link between some CVRF and kidney function? Can we see kidney function as a proxy for treatment efficiency meaning that even if CVRF seem controlled by strictly looking at corresponding direct markers, patients could still be at risk because kidney function is impaired?
Reviewer 2 Report
1. This is epidemiological research that covers the topic of the proportion of septuagenarians and octogenarians that is continually growing in Western countries. The study is of significant value and relevance.
2. The eponym AugUR 3 study is described in German, it should be translated into English, as in article reference 26.
3. The objective is clear, however when it comes to the method, reference 26 is cited. I believe that the reader may not be interested in reading reference 26, so it would be important to have a summary, a brief summary so that the article itself can be read by itself.
4. The discussion should focus more on the objective and results of the chosen population sample to reinforce or mitigate the conclusions.
Round 2
Reviewer 2 Report
The authors accepted all suggestions—excellent last version.
Congrats